# Paradoxical Induction of ALOX15/15B by Cortisol in Human Amnion Fibroblasts: Implications for Inflammatory Responses of the Fetal Membranes at Parturition

**DOI:** 10.3390/ijms241310881

**Published:** 2023-06-29

**Authors:** Fan Zhang, Jiang-Wen Lu, Wen-Jia Lei, Meng-Die Li, Fan Pan, Yi-Kai Lin, Wang-Sheng Wang, Kang Sun

**Affiliations:** 1Center for Reproductive Medicine, Ren Ji Hospital, School of Medicine, Shanghai Jiao Tong University, Shanghai 200135, China; zhangfan_0120@163.com (F.Z.); lujiangwen1110@163.com (J.-W.L.); leiwenjia257248@163.com (W.-J.L.); mengdieli777@163.com (M.-D.L.); panfan_1996@163.com (F.P.); ilinyikai@163.com (Y.-K.L.); 2Shanghai Key Laboratory for Assisted Reproduction and Reproductive Genetics, Shanghai 200135, China

**Keywords:** parturition, inflammation, ALOX15, 15(S)-HETE, PGE2, glucocorticoids

## Abstract

Inflammation of the fetal membranes is an indispensable event of parturition, with increasing prostaglandin E2 (PGE2) synthesis as one of the ultimate products that prime labor onset. In addition to PGE2, the fetal membranes also boast a large capacity for cortisol regeneration. It is intriguing how increased PGE2 synthesis is achieved in the presence of increasing amounts of classical anti-inflammatory glucocorticoids in the fetal membranes at parturition. 15(S)-hydroxyeicosatetraenoic acid (15(S)-HETE) synthesized by lipoxygenase 15/15B (ALOX15/15B) has been shown to enhance inflammation-induced PGE2 synthesis in amnion fibroblasts. Here, we examined whether glucocorticoids could induce ALOX15/15B expression and 15(S)-HETE production to promote PGE2 synthesis in amnion fibroblasts at parturition. We found that cortisol and 15(S)-HETE abundance increased parallelly in the amnion at parturition. Cortisol induced ALOX15/15B expression and 15(S)-HETE production paradoxically in amnion fibroblasts. Mechanism study revealed that this paradoxical induction was mediated by p300-mediated histone acetylation and interaction of glucocorticoid receptor with transcription factors CREB and STAT3. Conclusively, cortisol regenerated in the fetal membranes can paradoxically induce ALOX15/15B expression and 15(S)-HETE production in human amnion fibroblasts, which may further assist in the induction of PGE2 synthesis in the inflammatory responses of the fetal membranes for parturition.

## 1. Introduction

Inflammation of the fetal membranes plays a pivotal role in human parturition [1,2,3,4]. Inflammation not only weakens the tensile strength of the membranes culminating in membrane rupture [5,6,7], but also amplifies the synthesis of prostaglandins, particularly prostaglandin E2 (PGE2) which is derived primarily from the amnion layer of the fetal membranes in pregnancy [8,9,10,11,12,13]. Together with prostaglandin F2α (PGF2α), PGE2 has been recognized as one of the crucial ultimate common mediators of labor onset in virtually all species with involvement in cervical ripening, myometrial contraction, and membrane activation [8,14,15,16]. Interestingly, in addition to being a PGE2 reservoir, the human fetal membranes also boast the largest capacity for cortisol regeneration among fetal tissues in pregnancy [17,18,19]. It is known that this capacity is mediated through the reductase activity of 11β-hydroxysteroid dehydrogenase 1 (11β-HSD1) [19], which increases steadily with gestational age [20,21], and can be induced synergistically by cortisol and proinflammatory cytokines in inflammation [22,23,24]. Cortisol is a well-recognized classical anti-inflammatory hormone with inhibition of prostaglandin synthesis comprising one of its major anti-inflammatory actions [25,26,27]. It is intriguing how the abundant PGE2 synthesis is achieved in the presence of increasing cortisol regeneration in the inflammatory responses of the fetal membranes at parturition. We and others have previously demonstrated that glucocorticoids can paradoxically induce the expression of cyclooxygenase 2 (COX-2), the rate-limiting enzyme in prostaglandin synthesis [28], in human amnion fibroblasts [13,29,30,31,32,33,34], which may explain at least in part for the paradoxically simultaneous increases in PGE2 synthesis and cortisol regeneration in the inflammatory responses of the fetal membranes at parturition.

Recently, we identified another important pathway underlying ample PGE2 synthesis in the inflammatory responses of the fetal membranes at parturition [35]. We found that 15(S)-hydroxyeicosatetraenoic acid (15(S)-HETE), formed from arachidonic acid through the lipoxygenase 15 (ALOX15) and 15B (ALOX15B) pathway, can greatly bolster pro-inflammatory mediators-induced COX-2 expression and PGE2 synthesis in human amnion fibroblasts [35]. In turn, PGE2 can also induce ALOX15/15B expression and 15(S)-HETE synthesis through the cAMP/PKA signaling pathway coupled to its EP2 receptor, thereby forming a feed-forward loop between 15(S)-HETE and PGE2 synthesis in amnion fibroblasts at parturition [35]. In addition to the induction of prostaglandin synthesis, 15(S)-HETE has also been shown to be involved in a number of other inflammatory reactions under pathological conditions [36,37,38,39]. Given the pro-inflammatory role of 15(S)-HETE, expectedly, glucocorticoids normally inhibit ALOX15/15B expression and 15(S)-HETE synthesis in most non-gestational tissues [40,41,42,43,44]. However, in consideration of the paradoxical induction of COX-2 expression and PGE2 synthesis by cortisol [13,29,30,31,32,33,34] and the induction of ALOX15/15B expression and 15(S)-HETE synthesis by PGE2 in human amnion fibroblasts [35], it is tempting to speculate that cortisol may induce rather than inhibit ALOX15/15B expression and 15(S)-HETE synthesis indirectly via PGE2 in human amnion fibroblasts, so that the feed-forward loop between 15(S)-HETE and PGE2 synthesis can be reinforced in the presence of increasing cortisol regeneration in amnion fibroblasts in the inflammatory responses of the fetal membranes at parturition.

Histone modifications can alter chromatin structural organization to control the accessibility of transcription factors to gene promoter [45,46]. Among histone modifications, acetylation by histone acetyltransferases (HATs) is typically known as a modification to open up the compacted chromatin structure to allow the access of transcription factors to the gene promoter for the simulation of gene expression [47,48]. P300 is such a well-recognized HAT, which acts not only as a histone acetyltransferase [49] but also as a transcriptional adaptor for multiple transcription factors [50,51,52,53,54]. Bioinformatics analysis revealed that both *ALOX15* and *15B* promoters bear putative p300 binding sites. Moreover, cortisol has been shown to be capable of inducing p300 expression in human amnion fibroblasts [51], suggesting that p300 may be involved in the paradoxical induction of *ALOX15* and *15B* by cortisol in human amnion fibroblasts. In addition to p300 binding sites, *ALOX15* and *15B* promoters also bear multiple binding sites for glucocorticoids receptor (GR), signal transducer and activator of transcription 3 (STAT3) and cAMP response element-binding protein (CREB), all of which have been shown to be downstream effectors of cortisol in human amnion fibroblasts [32,33]. Unlike the direct activation of GR by cortisol, the activation of STAT3 and CREB by cortisol appears to be secondary to its induction of PGE2 synthesis in human amnion fibroblasts [33,34]. In turn, PGE2 activates STAT3 and CREB by phosphorylation through the cAMP/PKA pathway coupled with its EP2 receptor [55]. Thus, we hypothesized that cortisol might induce *ALOX15/15B* expression and 15(S)-HETE synthesis paradoxically in human amnion fibroblasts through p300-mediated histone acetylation as well as GR, STAT3, and CREB following activation by PGE2 via the cAMP/PKA pathway coupled with its EP2 receptor, so that the feed-forward loop between 15(S)-HETE and PGE2 synthesis can be reinforced in amnion fibroblasts by cortisol regeneration to ensure adequate 15(S)-HETE and PGE2 synthesis for parturition. Here, we examined the hypothesis in cultured primary human amnion fibroblasts and further verified the participating factors in human amnion collected from spontaneous labor.

## 2. Results

### 2.1. Parallel Increases in 15(S)-HETE and Cortisol in Human Amnion at Parturition

Firstly, we examined whether the abundance of 15(S)-HETE increased along with that of cortisol in the human amnion tissue in spontaneous labor at term. Enzyme-linked immunosorbent assay (ELISA) showed that the abundance of both cortisol and 15(S)-HETE increased significantly in the amnion tissue in the TL (term labor) group when compared with that in the TNL (term non-labor) group (Figure 1A,B). Pearson correlation analysis showed a positive correlation (R = 0.81, *p* = 0.0005) between 15(S)-HETE and cortisol levels in the amnion tissue (Figure 1C), indicating that the increased synthesis of 15(S)-HETE might be associated with increased cortisol regeneration in the amnion.

### 2.2. Effect of Cortisol on ALOX15/15B Expression and 15(S)-HETE Synthesis in the Human Amnion

Analysis of our previous transcriptomic data (NCBI GEO accession GES166320) obtained from human amnion fibroblasts treated with and without cortisol (1 μM; 24 h) [21] showed that *ALOX15* and *ALOX15B* mRNA but not the other members of the *ALOX* family (*ALOX5, ALOX12, ALOX12B, ALOXE3*) were significantly increased by cortisol treatment (1 μM; 24 h) (Figure 1D,E). Here, we confirmed the induction of ALOX15/15B expression by cortisol in human amnion fibroblasts with quantitative real-time polymerase chain reaction (qRT-PCR) and Western blotting. Both qRT-PCR and Western blotting showed that cortisol (0.01, 0.1, and 1 μM; 24 h) increased ALOX15/15B mRNA and protein abundance (Figure 1F–I) and 15(S)-HETE production (Figure 1J) in a concentration-dependent manner in human amnion fibroblasts. Immunofluorescence staining showed that the staining of ALOX15/15B was hardly detectable in human amnion fibroblasts in the absence of cortisol treatment but became rather intense after cortisol treatment (1 μM; 24 h) (Figure 1K). These results indicate that cortisol can induce ALOX15/15B expression and 15(S)-HETE synthesis paradoxically in human amnion fibroblasts at parturition.

### 2.3. Involvement of p300-Mediated Histone Acetylation in the Induction of ALOX15/15B by Cortisol in Human Amnion Fibroblasts

In silico analysis of *ALOX15* and *ALOX15B* promoters (−650 bp~+250 bp) revealed multiple p300 binding sites (Appendix A). Cortisol (1 μM; 24 h) treatment of human amnion fibroblasts increased p300 mRNA and protein abundance significantly (Figure 2A). Inhibition of p300 with C646 (10 μM) blocked the induction of ALOX15/15B mRNA and protein by cortisol (1 μM; 24 h) (Figure 2B,C). Chromatin immunoprecipitation (ChIP) assay showed that cortisol treatment (1 μM; 12 h) significantly increased the enrichments of p300 and acetylated H3K27 (H3K27ac) at *ALOX15* (−284 bp~−89 bp) and *ALOX15B* (+13 bp~+179 bp) promoters in human amnion fibroblasts (Figure 2D–G). These data suggest that p300-mediated H3K27ac are involved in the induction of ALOX15 and 15B expression by cortisol in human amnion fibroblasts.

### 2.4. Role of GR, CREB, and STAT3 in the Induction of ALOX15/15B by Cortisol in Human Amnion Fibroblasts

In silico analysis of *ALOX15* and *15B* promoters (−650 bp ~ +250 bp) revealed the presence of binding sites for GR, CREB, and STAT3 adjacent to the binding site for p300 (Appendix A). Subsequent studies with antagonists showed that the induction of ALOX15/15B mRNA and protein by cortisol (1 μM; 24 h) was abolished by either RU486 (1 μM) (Figure 3A–D) or S3I-201 (10 μM) (Figure 3E–H) or 666-15 (10 μM) (Figure 3I–L), the respective GR, STAT3, and CREB antagonists, indicating the participation of GR, STAT3, and CREB in the induction of ALOX15 and 15B by cortisol. The notion was further endorsed by ChIP and co-immunoprecipitation (CoIP) assays. ChIP assay showed increased enrichments of GR, CREB, and STAT3 at *ALOX15* (−284 bp~−89 bp) and *ALOX15B* (+13 bp~+179 bp) promoters following cortisol treatment (1 μM; 12 h) in human amnion fibroblasts (Figure 4A–D). CoIP assay revealed that p300, GR, CREB, and STAT3 were in the same nuclear protein complex upon cortisol treatment (1 μM; 12 h) (Figure 4E). These data suggest that the induction of p300-mediated H3K27ac by cortisol may open up the chromatin structure to allow the access of GR, CREB, and STAT3 to *ALOX15* and *ALOX15B* promoters for their interaction with p300 to induce *ALOX15/15B* expression in human amnion fibroblasts.

### 2.5. Role of the cAMP/PKA Pathway Coupled with the EP2 Receptor of PGE2 in the Induction of ALOX15/15B Expression by Cortisol in Human Amnion Fibroblasts

Our previous studies have demonstrated that the cAMP/PKA pathway coupled with the EP2 receptor is involved in the activation of CREB and STAT3 in the induction of COX-2 and ALOX15/15B expression by PGE2 in human amnion fibroblasts [32,33,35,55]. Thus, we speculated that the activation of CREB and STAT3 and induction of ALOX15/15B expression by cortisol were secondary to its induction of PGE2 production. Thus, blocking the cAMP/PKA pathway coupled with the EP2 receptor of PGE2 should also abolish the induction of ALOX15/15B by cortisol. The notion was supported by the findings that the induction of CREB and STAT3 phosphorylation (3 h), as well as ALOX15/15B expression (24 h) by cortisol (1 μM), could be blocked by either EP2 receptor antagonist PF-04418948 (PF; 10 μM) (Figure 5A–F) or PKA inhibitor PKI 14-22 amide (PKI; 5 μM) (Figure 5G–L) in human amnion fibroblasts.

### 2.6. Role of the cAMP/PKA Pathway Coupled with the EP2 Receptor of PGE2 in the Induction of p300 Expression by Cortisol in Human Amnion Fibroblasts

In silico analysis of the *EP300* promoter (−640 bp ~ +200 bp) revealed multiple binding sites for CREB but not for GR or STAT3 (Appendix A), indicating that CREB may also be involved in the regulation of p300 expression by cortisol. This notion was supported by the findings that the induction of *EP300* mRNA expression by cortisol (1 μM; 24 h) could be blocked by CREB inhibitor 666-15 (10 μM) but not by STAT3 inhibitor S3I-201 (10 μM) (Figure 6A,B). ChIP assay showed that cortisol treatment (1 μM; 12 h) significantly increased the enrichment of CREB at *EP300* gene promoter (−613 bp ~ −388 bp) in human amnion fibroblasts (Figure 6C,D). Moreover, the induction of *EP300* expression by cortisol (1 μM; 24 h) could be blocked by either EP2 receptor antagonist PF-04418948 (PF; 10 μM) or PKA inhibitor PKI 14-22 amide (PKI; 5 μM) (Figure 6E,F) in human amnion fibroblasts. Consistently, PGE2 could also induce *EP300* expression in human amnion fibroblasts (Figure 6G), which was blocked by either EP2 receptor antagonist PF-04418948 (PF; 10 μM) or PKA inhibitor PKI 14-22 amide (PKI; 5 μM) (Figure 6H,I). These results indicate that cortisol induces *EP300* expression indirectly following its induction of PGE2 synthesis, which, in turn, induces *EP300* expression through activation of the cAMP/PKA/CREB pathway coupled with the EP2 receptor.

### 2.7. Correlation among ALOX15/15B, p300, p-STAT3 and p-CREB Abundance in the Human Amnion at Parturition

Based on the in vitro findings in human amnion fibroblasts, we proceeded to examine whether the abundance of ALOX15/15B, p300, phosphorylated CREB (p-CREB), and phosphorylated STAT3 (p-STAT3) was altered correspondingly in the human amnion at parturition. Western blotting showed that the abundance of ALOX15, ALOX15B, p300, p-CREB, and p-STAT3 all increased significantly in the amnion of the TL group when compared with the TNL group (Figure 7A–F). Correlation analysis showed that the abundance of p300, p-CREB, and p-STAT3 correlated positively with that of ALOX15 and ALOX15B (Figure 7G–L), and the abundance of p-CREB also correlated positively with that of p300 in the amnion (Figure 7M). These data lend further support for the role of p300, CREB, and STAT3 in the induction of ALOX15 and ALOX15B expression, and the role of CREB in the induction of p300 expression in the human amnion at parturition.

## 3. Discussion

Cortisol regeneration by 11β-HSD1 increases significantly towards the end of gestation and further increases at parturition in the fetal membranes [20,21,56]. Moreover, cortisol regeneration can be further enhanced in the inflammatory responses of the fetal membranes given the synergistic induction of 11β-HSD1 by cortisol and pro-inflammatory cytokines [22,23,24]**.** We and others have demonstrated that cortisol regenerated in the fetal membranes takes part not only in progesterone withdrawal via induction of 20α-hydroxysteroid dehydrogenase 1 expression [57] but also in membrane rupture via downregulation of lysyl oxidase and promotion of collagen decrosslinking and degradation [58,59,60,61] at parturition. Moreover, cortisol regenerated also participates in the upregulation of prostaglandin synthesis via paradoxical induction COX-2 expression in human amnion fibroblasts at parturition [13,29,32,33,55,62]. In the present study, we revealed another parturition-pertinent paradoxical effect of cortisol in human amnion fibroblasts, i.e., induction of ALOX15/15B expression and 15(S)-HETE synthesis, which apparently contradicts the classical inhibition of ALOX15/15B expression by glucocorticoids in most of non-gestational tissues [40,41,42,43,44]. We have previously demonstrated that 15(S)-HETE produced by ALOX15/15B participates in parturition by potentiating the induction of COX-2 expression and PGE2 synthesis by pro-inflammatory factors in amnion fibroblasts [35]. The paradoxical induction of ALOX15/15B and COX-2 expression by cortisol in human amnion fibroblasts can further explain how the feed-forward loop between PGE2 and 15(S)-HETE syntheses is set up in the presence of ample cortisol regeneration in the inflammatory responses of the fetal membranes at parturition.

Our previous studies have shown that the interaction of GR with phosphorylated CREB/STAT3 following activation by the cAMP/PKA pathway coupled with the EP2 receptor of PGE2 holds the key to the paradoxical induction of COX-2 expression by cortisol in human amnion fibroblasts [32,33,34]. Here, in the present study, we found that cortisol employed the same signaling pathway to induce ALOX15/15B expression in human amnion fibroblasts. In addition, we found that CREB activated in this pathway also participated in the induction of p300 expression by cortisol in human amnion fibroblasts. Since the induction of both ALOX15/15B and p300 expression by cortisol could be blocked by EP2 and PKA antagonists, we believe that activation of the cAMP/PKA pathway coupled with the EP2 receptor by PGE2 is a prerequisite for the paradoxical induction of ALOX15/15B and p300 expression by cortisol in human amnion fibroblasts. In other words, this paradoxical effect of cortisol on ALOX15/15B expression requires, as a matter of fact, the interaction of GR with CREB and STAT3 upon activation by PGE2 via its EP2 receptor. This notion was supported not only by the ChIP assay but also by the CoIP assay, which revealed GR in the same nuclear protein complex with CREB and STAT3 upon cortisol treatment in human amnion fibroblasts. However, this nuclear protein complex appears to be not necessary for the induction of p300 by cortisol because the *EP300* promoter carries only binding sites only for CREB but not for STAT3 and GR, and moreover, STAT3 inhibitor was unable to block the induction of *EP300* expression by cortisol. Our data indicate that a different pool of activated CREB which is independent of the identified GR/CREB/STAT3 complex should exist in the induction of p300 expression in amnion fibroblasts by cortisol. However, we are unclear whether the stimulation of p300 expression requires CREB to form a nuclear complex with other transcription factors by cortisol in human amnion fibroblasts. Of note, one of the limitations of this study is that only antagonists to the relevant molecules were used. Small interfering RNA-mediated specific knockdown of these molecules would lend further support for the identified signaling pathway.

P300 belongs to the p300/CBP (CREB-binding protein) superfamily [49]. P300 and CBP were originally identified as transcriptional coactivators on the basis of their interaction with phosphorylated CREB [63] and adenovirus E1A oncoprotein [64]. It is now recognized that p300/CBP controls gene transcription through at least two regulatory mechanisms [65,66]. One mechanism is to increase the target gene transcription by relaxing the chromatin structure through the intrinsic histone acetyltransferase activity of p300/CBP with subsequent recruitment of the basal transcriptional machinery including RNA polymerase II to the gene promoter [49,67]. The other mechanism is to function as a bridge or scaffold for the interaction of p300/CBP with differential transcription factors, thereby stabilizing the transcriptional complex and promoting transcriptional synergy [50,51,52,53,54]. Here, we demonstrated that p300 was involved in the induction of ALOX15/15B expression by cortisol in human amnion fibroblasts, which is consistent with previous studies showing the involvement of p300 in the regulation of ALOX15/15B expression in A549 lung epithelial cells and SH-SY5Y human neuroblastoma cells [68,69]. Several lines of evidence indicate that GR, CREB, and STAT3 are among the numerous transcription factors that can interact with p300 for transcriptional activation [51,53,54,70]. We also found that p300, GR, CREB, and STAT3 were in the same transcriptional complex that could be enriched at the *ALOX15* and *ALOX15B* gene promoters upon cortisol stimulation of human amnion fibroblasts, suggesting that p300 regulates ALOX15/15B expression not only through its intrinsic histone acetyltransferase but also through its bridging or scaffolding action bringing the involved transcription factors together. However, given the broad regulatory role of H3K27ac in gene expression, in addition to *ALOX15/15B*, a series of other genes pertinent to parturition may also be subject to the regulation by p300 in human amnion fibroblasts. Of interest, our previous study has shown that p300 is also involved in the induction of hexose-6-phosphate dehydrogenase (H6PD) in human amnion fibroblasts, which produces the cofactor NADPH for the cortisol-regenerating activity of 11β-HSD1 [51]. This effect of p300 on H6PD expression and NADPH production would further increase cortisol regeneration to enhance its parturition-pertinent effects in the fetal membranes.

In conclusion, we have demonstrated in this study that cortisol regenerated in the human fetal membranes can paradoxically induce ALOX15/15B expression and 15(S)-HETE production in human amnion fibroblasts, which is dependent on the cAMP/PKA signaling pathway activated by PGE2 via its EP2 receptor (Figure 8). These findings may further explain how the feed-forward loop between 15(S)-HETE and PGE2 synthesis is set up in the presence of ample cortisol regeneration in the inflammatory responses of the fetal membranes at parturition. The intervention of this feed-forward loop may be of therapeutic value in the treatment of preterm birth.

## 4. Materials and Methods

### 4.1. Collection of Human Fetal Membranes

Human fetal membranes were obtained from pregnancies at term (38–40 weeks) following spontaneous labor (designated as term labor, TL) and elective cesarean section in the absence of labor (designated as term non-labor, TNL) with written informed consent under a protocol approved by the Ethics Committee of Ren Ji Hospital, Shanghai Jiao Tong University School of Medicine. Pregnancies with complications such as preeclampsia, fetal growth restriction, gestational diabetes, and chorioamnionitis were excluded from the study. Upon deliveries, the amnion layer at the spontaneous (TL) or artificial (TNL) rupture sites was immediately peeled off the fetal membranes, and snap-frozen in liquid nitrogen for later processing for determination of cortisol, 15(S)-HETE, ALOX15, ALOX15B, CREB, p-CREB, STAT3, p-STAT3, and p300 in the amnion tissue as described below. The entire amnion layer of the reflected membranes obtained from TNL was used for isolation of amnion fibroblast cells. Detailed information on recruited women is listed in Appendix A.

### 4.2. Measurement of Cortisol and 15(S)-HETE in Human Amnion with ELISA

Cortisol and 15(S)-HETE in the amnion tissue were extracted from the snap-frozen tissue with ethyl acetate after homogenization in PBS. Upon evaporation of ethyl acetate, the extract was reconstituted in the assay buffer for measurements of cortisol and 15(S)-HETE with ELISA kits (both from Cayman Chemical, Ann Arbor, MI, USA) following the protocol provided by the manufacturer.

### 4.3. Isolation and Culture of Human Amnion Fibroblasts

The entire amnion from the reflected membranes obtained from TNL was used for isolation of amnion fibroblasts as described previously [71]. Briefly, the amnion tissue was digested twice with 0.125% trypsin (Life Technologies Inc., Grand Island, NY, USA), and then washed vigorously with normal saline to remove epithelial cells. The remaining amnion mesenchymal tissue was further digested with 0.1% collagenase (Sigma, St. Louis, MO, USA) for isolation of fibroblasts. The isolated amnion fibroblasts were collected by centrifugation and resuspended in Dulbecco’s modified eagle medium (DMEM) containing 10% fetal bovine serum (FBS) and antibiotics (all from Thermo Fisher Scientific, Waltham, MA, USA) for culture at 37 °C in 5% CO_2_/95% air. This method of amnion cell isolation yields high purity of fibroblasts, which have been previously characterized with immunofluorescence staining, and the majority of cells are positive for the mesenchymal marker vimentin but not for the epithelial marker cytokeratin-7 or the immune cell marker CD45 [58].

### 4.4. Treatment of Human Amnion Fibroblasts

Three days after plating, cultured amnion fibroblasts were treated with the following reagents in phenol red/FBS-free culture medium for 24 h unless specified. The dose-dependent effect of cortisol on ALOX15 and ALOX15B abundance were examined by treating the cells with 0.01, 0.1 and 1 μM cortisol. The effect of cortisol on p300 expression was examined by treating the cells with 1 μM cortisol. To investigate the role of p300 in the regulation of ALOX15/15B expression by cortisol, fibroblasts were treated with cortisol (1 μM) in the presence or absence of p300 inhibitor C646 (10 μM; Selleck, Houston, TX, USA). To study whether the effect of cortisol is dependent on GR, fibroblasts were treated with 1 μM cortisol in the presence or absence of GR antagonist RU486 (1 μM; Sigma). To investigate the role of CREB or STAT3 in the regulation of ALOX15/15B or p300 expression by cortisol, fibroblasts were treated with 1 μM cortisol in the presence or absence of CREB inhibitor 666-15 (10 μM; Selleck) or STAT3 inhibitor S3I-201 (10 μM; Selleck). To explore whether cortisol induced the phosphorylation of CREB and STAT3 or the expression of ALOX15, ALOX15B, and p300 through the cAMP/PKA pathway coupled to the EP2 receptor of PGE2, amnion fibroblasts were treated with 1 μM cortisol in the presence or absence of EP2 receptor antagonist PF-04418948 (PF; 10 μM; Selleck) or PKA inhibitor PKI 14-22 amide (PKI; 5 μM; Selleck) for 3 h for phosphorylation study, and for 24 h for gene expression study. To study whether PGE2 can induce p300 expression through the cAMP/PKA pathway coupled to its EP2 receptor, amnion fibroblasts were treated with PGE2 (1 μM; 6 h) in the presence or absence of EP2 receptor antagonist PF-04418948 (PF; 10 μM; Selleck) or PKA inhibitor PKI 14-22 amide (PKI; 5 μM; Selleck). To examine the effect of cortisol on the enrichments of p300, H3K27ac, GR, CREB, and STAT3 at *ALOX15* and *ALOX15B* promoters or the enrichment of CREB at the *EP300* promoter, fibroblasts were treated with 1 μM cortisol for 12 h. To examine whether p300 could form a complex with GR, STAT3, and CREB, fibroblasts were treated with 1 μM cortisol for 12 h. All the inhibitors were added 1 h before cortisol treatment. After treatment, the conditioned culture medium was collected for the measurement of 15(S)-HETE with an ELISA kit (Cayman Chemical) according to the protocol provided by the manufacturer, and the cells were processed for extraction of total RNA, cellular or nuclear protein.

### 4.5. Extraction of RNA and Analysis with qRT-PCR

Total RNA was extracted from the treated cells using a commercial total RNA isolation Kit (Foregene, Chengdu, China). After examination of RNA quality, reverse transcription was conducted using a Prime-Script RT Master Mix Kit (TaKaRa, Kyoto, Japan). The amount of *ALOX15*, *ALOX15B,* and *EP300* mRNA was determined with qRT-PCR using the above reverse-transcribed cDNA and power SYBR^®^ Premix Ex Taq™ (TaKaRa). Housekeeping gene *GAPDH* (encoding glyceraldehyde 3-phosphate dehydrogenase) was amplified in parallel as an internal control. The relative mRNA abundance was quantitated using the 2^−∆∆Ct^ method. The primer sequences used for qRT-PCR are illustrated in Appendix A.

### 4.6. Extraction of Protein and Analysis with Western Blotting

Total cellular protein was extracted from the snap-frozen amnion tissue or treated cells with an ice-cold radioimmunoprecipitation assay (RIPA) lysis buffer (Active Motif, Carlsbad, CA, USA) containing inhibitors for protease and phosphatase (Roche, Indianapolis, IN, USA). Nuclear protein was extracted from treated cells using a Nuclear Extract Kit (Active Motif) according to the protocol provided by the manufacturer. After determination of protein concentration with the Bradford method, the protein abundance of ALOX15, ALOX15B, p300, total CREB, p-CREB (Ser^133^), total STAT3, p-STAT3 (Tyr^705^) was determined following a standard Western blotting protocol. Briefly, 30 μg of total protein was electrophoresed in a sodium dodecyl sulfate (SDS)-polyacrylamide gel. After transferring to a nitrocellulose membrane blot, the blot was blocked with 5% non-fat milk, and incubated with antibodies against ALOX15 (1:500; Thermo Fisher Scientific; #MA5-25853), ALOX15B (1:500; Thermo Fisher Scientific; #PA5-97456), p300 (1:200; Abcam, Cambridge, U.K.; #ab14984), total CREB (1:1000; Cell Signaling, Danvers, MA, USA; #9104S), p-CREB (Ser^133^) (1:1000; Cell Signaling; #9198S), total STAT3 (1:1000; Cell Signaling; #9139S) and p-STAT3 (Tyr^705^) (1:1000; Cell Signaling; #9145S) overnight at 4 °C, followed by incubation with appropriate secondary antibodies conjugated with horseradish peroxidase (Proteintech, Wuhan, China). Peroxidase activity was developed with a chemiluminescence detection system (Millipore, Billerica, MA, USA), and visualized using a G-Box chemiluminescence image capture system (Syngene, Cambridge, UK). Internal loading control was performed by probing the blot with the GAPDH (1:10,000; Proteintech; #60004-1) antibody for cellular protein or the Lamin A/C antibody (1:500; Cell signaling; #4777S) for nuclear protein. The ratio of band intensities of ALOX15 and ALOX15B to that of GAPDH was used to indicate ALOX15 and ALOX15B protein abundance. The ratio of p300 band density over that of Lamin A/C was used to indicate p300 protein abundance in the nucleus. The ratio of phosphorylated CREB or STAT3 band density over that of total CREB or STAT3 was used to indicate the abundance of phosphorylated CREB or STAT3. All the antibody information is illustrated in Appendix A.

### 4.7. Immunofluorescence Staining

To examine the induction of ALOX15/15B by cortisol with immunofluorescence staining, amnion fibroblasts with or without cortisol (1 μM; 24 h) treatment were fixed with 4% paraformaldehyde, and permeabilized with 0.4% Triton X-100 for immunofluorescence staining. Briefly, after blocking with normal serum, the cells were incubated with antibodies against ALOX15 (Thermo Fisher Scientific; #MA5-25853) or ALOX15B (Thermo Fisher Scientific; #PA5-97456) at 1:100 dilution, followed by incubation with secondary antibodies conjugated with Alexa Fluor 488 (green color) or Alexa Fluor 594 (red color) (Proteintech) respectively. Nuclei were visualized with DAPI (1 μg/mL; blue color) staining. A fluorescence microscope (Zeiss, Jena, Germany) was used to observe immunofluorescence staining.

### 4.8. ChIP Assay

The enrichments of p300, H3K27ac, GR, CREB, and STAT3 at *ALOX15* and *ALOX15B* promoters, or the enrichment of CREB at *EP300* promoter following cortisol treatment were determined with ChIP assay as described previously [21]. Briefly, after crosslinking with 1% formaldehyde, amnion fibroblasts were lysed with SDS lysis buffer containing a protease inhibitor cocktail. The lysed cells were sonicated to shear the chromatin DNA to optimal size of around 500 bp, which was then immunoprecipitated with antibodies against p300 (Abcam; #ab14984), H3K27ac (Abcam; #ab4729), GR (Cell Signaling; #12041S), total CREB (Cell Signaling; #9104S) or total STAT3 (Cell Signaling; #9139S). An equal amount of pre-immune IgG served as negative control. Sheared DNA without immunoprecipitation served as input control. The immunoprecipitate was pulled down with Magna ChIP Protein A+G agarose Magnetic Beads (Millipore) on a magnetic stand. After reverse cross-linking, protein and RNA were removed by digestion with proteinase K and ribonuclease, respectively. The sheared DNA was extracted using a DNA purification kit (Cwbiotech, Beijing, China) for subsequent qRT-PCR with primers aligning the putative binding sites of p300, H3K27, GR, CREB, or STAT3 at *ALOX15* and *ALOX15B* promoters (Appendix A) or aligning the putative binding site of CREB at *EP300* promoter (Appendix A). The primer sequences for ChIP are illustrated in Appendix A. The ratio of DNA precipitated by p300, H3K27ac, GR, CREB, or STAT3 antibodies over that of input control was calculated as a measure of bound p300, H3K27ac, GR, CREB, and STAT3.

### 4.9. CoIP Assay

CoIP assay was carried out to examine whether p300, STAT3, GR, and CREB could form a complex after cortisol treatment. Nuclear protein was extracted from cells treated with or without cortisol (1 μM; 12 h) using a Nuclear Extract Kit (Active Motif). After determination of protein concentration, 10 μg nuclear protein was incubated with 1:50 dilution of rabbit antibody against p300 (Abcam; #ab14984), or pre-immune rabbit IgG as negative control overnight at 4 °C. Then, protein A agarose beads were added for incubation with the above reaction mixture for 1 h on ice to pull down the antibody/antigen complex. The precipitated antibody/antigen/agarose complex was washed adequately, and then denatured in Western blot loading buffer at 95 °C for subsequent detection of STAT3, GR, and CREB with Western blotting. The bands were visualized in the same way as Western blotting.

### 4.10. Statistical Analysis

All data are presented as means ± SEM. The number of repeated experiments in each study was separate experiments using amnion samples prepared from independent pregnant women. After normality testing with the Shapiro–Wilk test, paired or unpaired Student’s *t*-test or Mann–Whitney U test was used to compare two groups. One-way ANOVA followed by Newman–Keuls multiple comparisons test was performed when assessing the differences among multiple groups. Pearson correlation analysis was performed to analyze the correlation between groups. Statistical significance was defined as *p* < 0.05.

Abbreviations: 15(S)-HETE, 15(S)-hydroxyeicosatetraenoic acid; LOX15/15B, lipoxygenase 15/15B; PGE2, prostaglandin E2; 11β-HSD1, 11β-hydroxysteroid dehydrogenase I; COX-2, cyclooxygenase 2; GR, glucocorticoid receptor; STAT3, signal transducer and activator of transcription 3; CREB, cAMP response element-binding protein; HATs, histone acetyltransferases; ELISA, enzyme-linked immunosorbent assay; TL, term labor; TNL, term non-labor; qRT-PCR; quantitative real-time polymerase chain reaction; ChIP, chromatin immunoprecipitation; CoIP, co-immunoprecipitation; CBP, CREB-binding protein; H3K27ac; acetylated H3K27; H6PD, hexose-6-phosphate dehydrogenase; DEME, Dulbecco’s modified eagle medium; FBS, fetal bovine serum; SDS, sodium dodecyl sulfate.

## Figures and Tables

**Figure 1 ijms-24-10881-f001:**
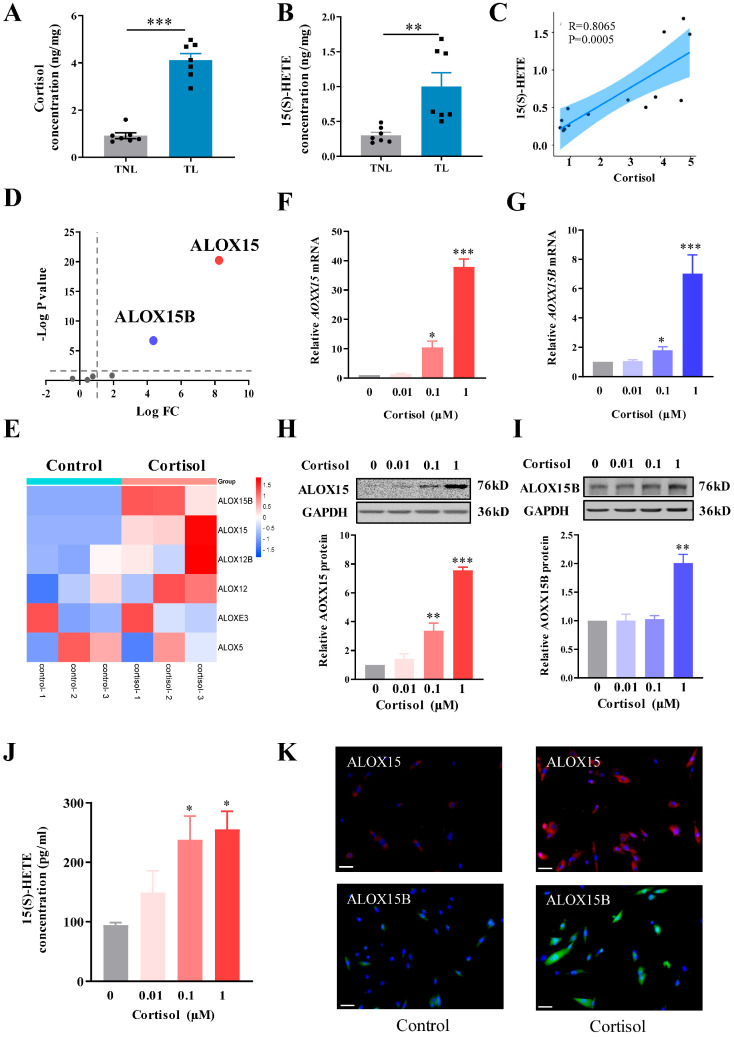
**Effect of cortisol on ALOX15 and ALOX15B expression in human amnion fibroblasts.** (**A**,**B**) Abundance of cortisol (**A**) and 15(S)-HETE (**B**) in the human amnion obtained from deliveries at term with labor (TL; n = 7) and without labor (TNL; n = 7). (**C**) Pearson analysis showing the positive correlation between 15(S)-HETE and cortisol levels in the human amnion (n = 14). (**D**) Scatter plot of the transcriptomic sequencing data displaying the fold change (FC) in *ALOX* family members in human amnion fibroblasts treated with or without cortisol (1 µM; 24 h). Y-axis represents −log (*p*-value) and X-axis represents log (FC). n = 3. (**E**) Heatmap of the transcriptomic sequencing data displaying the mRNA abundance of *ALOX* family members in human amnion fibroblasts with and without cortisol (1 µM; 24 h) treatment. Blue to red represents expression levels from low to high. n = 3. (**F**–**I**) Concentration-dependent induction of ALOX15 and ALOX15B mRNA (F, n = 6; G, n = 4) and protein (H, n = 3; I, n = 4) expression by cortisol (0.01, 0.1 and 1 µM; 24 h) in human amnion fibroblasts. (**J**) Concentration-dependent induction of 15(S)-HETE production by cortisol (0.01, 0.1, and 1 µM; 24 h) in human amnion fibroblasts. n = 3. (**K**) Representative images of immunofluorescence staining of ALOX15 (red) and ALOX15B (green) in human amnion fibroblasts with or without cortisol (1 µM; 24 h) treatment. Nuclei were stained with DAPI (blue). Scale bars, 50 μm. n = 3. Data are represented as means + SEM. Statistical analysis was performed with unpaired Student’s *t*-test (**A**,**B**) or one-way ANOVA test followed by Newman–Keuls multiple comparisons test (**F**–**I**,**J**). * *p* < 0.05, ** *p* < 0.01, *** *p* < 0.001 vs. TNL or control group.

**Figure 2 ijms-24-10881-f002:**
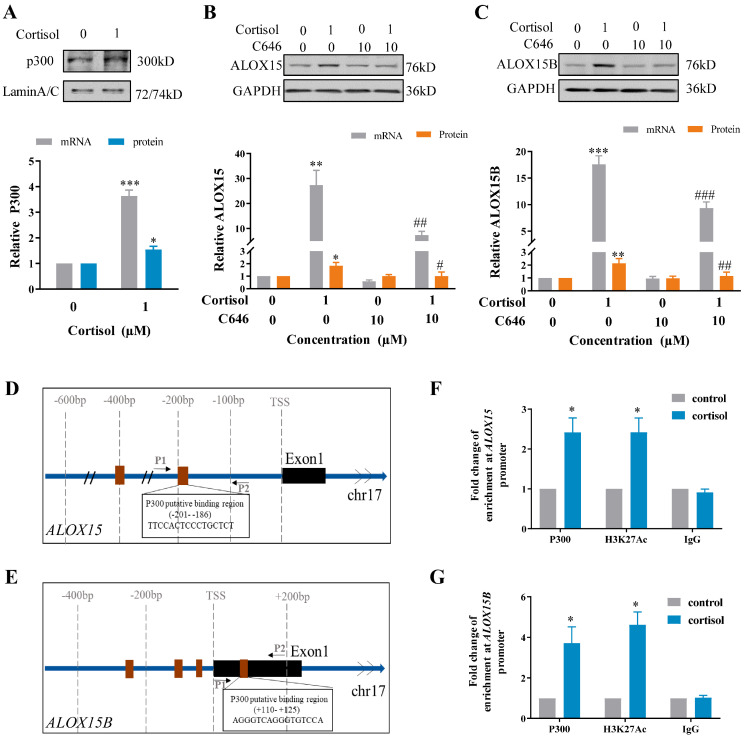
Involvement of p300 in the induction of ALOX15 and ALOX15B expression by cortisol in human amnion fibroblasts. (**A**) Induction of p300 mRNA (n = 3) and protein (n = 3) expression by cortisol (1 µM; 24 h). (**B**,**C**) Blockade of cortisol (1 µM; 24 h)-induced ALOX15 and ALOX15B mRNA (n = 4) and protein (n = 4) expression by p300 inhibitor C646 (10 µM). (**D**,**E**) Diagrams showing the putative binding sites of p300 in *ALOX15* (**D**) and *ALOX15B* (**E**) gene promoters. Brown box indicates p300 binding sites. Arrows indicate primer aligning positions in ChIP assay. P1, forward primer; P2, reverse primer; TSS, transcription start site. (**F**,**G**) ChIP assay showing enrichments of p300 and H3K27ac at *ALOX15* (**F**) and *ALOX15B* (**G**) gene promoters in human amnion fibroblasts with cortisol treatment (1 μM; 12 h). n = 4. IgG served as negative control. Statistical analysis was performed with paired Student’s *t*-test (**A**,**F**,**G**) or one-way ANOVA test followed by Newman–Keuls multiple comparisons test (**B**,**C**). Top panels of A−C are the representative immunoblots. * *p* < 0.05, ** *p* < 0.01, *** *p* < 0.001 vs. control group. # *p* < 0.05, ## *p* < 0.01, ### *p* < 0.001 vs. cortisol-treated group.

**Figure 3 ijms-24-10881-f003:**
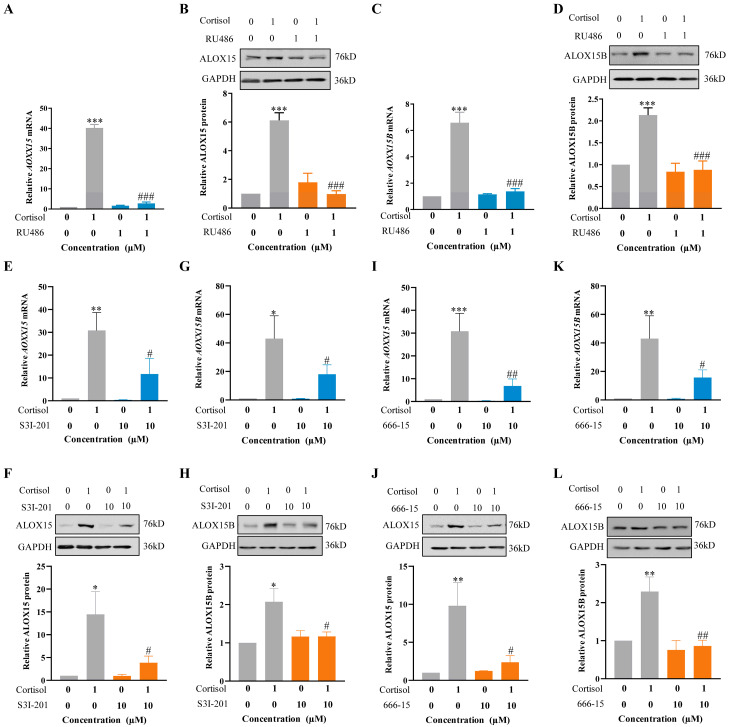
Involvement of GR, CREB, and STAT3 in the induction of ALOX15 and ALOX15B expression by cortisol in human amnion fibroblasts. (**A**–**D**) Blockade of cortisol (1 µM; 24 h)-induced ALOX15 and ALOX15B mRNA (**A**,**C**, n = 4) and protein (**B**,**D**, n = 4) expression by GR antagonist RU486 (1 µM). (**E**–**H**) Blockade of cortisol (1 µM; 24 h)-induced ALOX15 and ALOX15B mRNA (**E**,**G**, n = 4) and protein (**F**,**H**, n = 4) expression by STAT3 inhibitor S3I-201 (10 μM). (**I**–**L**) Blockade of cortisol (1 µM; 24 h)-induced ALOX15 and ALOX15B mRNA (**I**,**K**, n = 4) and protein (**J**,**L**, n = 4) expression by CREB inhibitor 666-15 (10 μM). Data are represented as means + SEM. Statistical analysis was performed with one-way ANOVA test followed by Newman–Keuls multiple comparisons test. Top panels of B, D, F, H, J, and L are the representative immunoblots. * *p* < 0.05, ** *p* < 0.01, *** *p* < 0.001 vs. control group. # *p* < 0.05, ## *p* < 0.01, ### *p* < 0.001 vs. cortisol-treated group.

**Figure 4 ijms-24-10881-f004:**
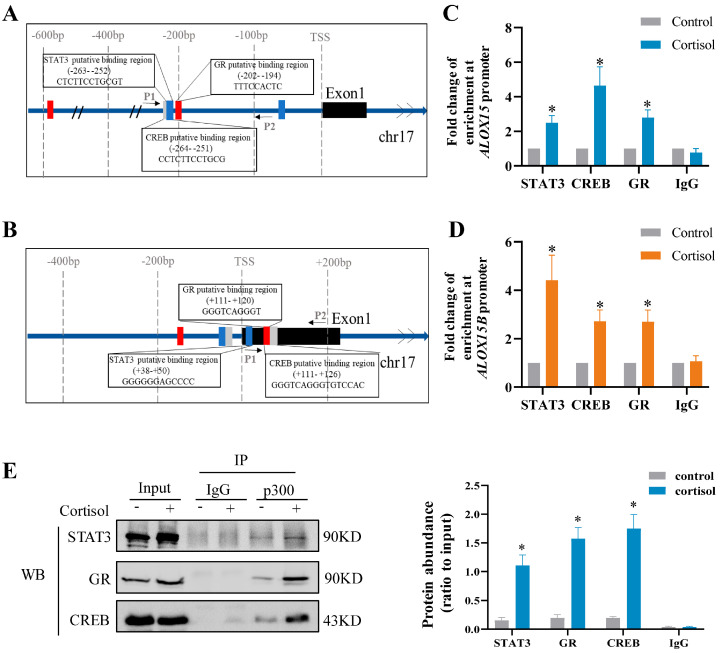
Increased enrichments of GR, CREB, and STAT3 at *ALOX15* and *ALOX15B* gene promoters in human amnion fibroblasts by cortisol treatment. (**A**,**B**) Diagrams showing the putative binding sites for GR, STAT3, and CREB in *ALOX15* (**A**) and *ALOX15B* (**B**) gene promoters. Red box indicates GR binding sites, blue box indicates STAT3 binding sites, and gray box indicates CREB binding sites. Arrows indicate primer aligning positions in ChIP assay. P1, forward primer; P2, reverse primer; TSS, transcription start site. (**C**,**D**) ChIP assay showing the enrichments of GR, STAT3, and CREB at *ALOX15* (**C**) and *ALOX15B* (**D**) gene promoters in human amnion fibroblasts upon cortisol treatment (1 μM; 12 h). n = 4. IgG served as negative control. (**E**) CoIP assay showed the interaction of p300 with STAT3, CREB, and GR in human amnion fibroblasts with cortisol (1 µM; 12 h) treatment. Input and IgG served as positive and negative control, respectively. n = 3. Data are represented as means + SEM. Statistical analysis was performed with paired Student’s *t*-test. * *p* < 0.05 vs. control group.

**Figure 5 ijms-24-10881-f005:**
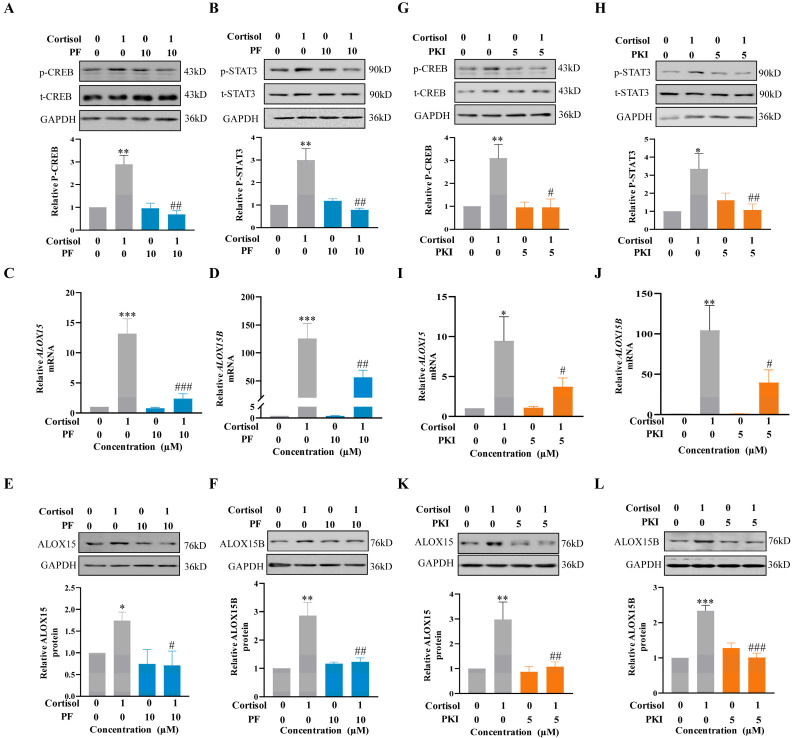
Role of PGE2-EP2-PKA pathway in the induction of CREB and STAT3 phosphorylation, and ALOX15/15B expression by cortisol in human amnion fibroblasts. (**A**–**F**) Blockade of cortisol (1 µM)-induced phosphorylation of CREB (**A**, n = 3) and STAT3 (**B**, n = 3) and increases in ALOX15 and ALOX15B mRNA (**C**,**D**, n = 4) and protein (**E**,**F**, n = 3) abundance by EP2 receptor antagonist PF-04418948 (PF; 10 μM). (**G**–**L**) Blockade of cortisol (1 µM)-induced phosphorylation of CREB (**G**, n = 3) and STAT3 (**H**, n = 3) and increases in ALOX15 and ALOX15B mRNA (**I**,**J**, n = 4) and protein abundance (**K**,**L**, n = 4) by PKA inhibitor PKI 14-22 amide (PKI; 5 μM). Data are represented as means + SEM. Statistical analysis was performed with one-way ANOVA test followed by Newman–Keuls multiple comparisons test. Top panels of (**A**,**B**,**E**,**F**,**G**,**H**,**K**,**L**) are the representative immunoblots. * *p* < 0.05, ** *p* < 0.01, *** *p* < 0.001 vs. control group. # *p* < 0.05, ## *p* < 0.01, ### *p* < 0.001 vs. cortisol-treated group.

**Figure 6 ijms-24-10881-f006:**
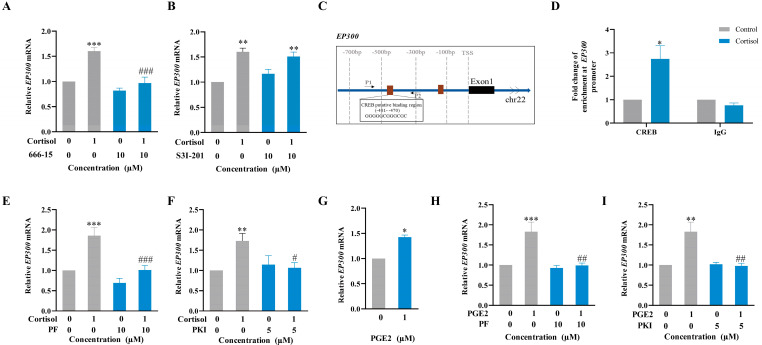
Involvement of CREB but not STAT3 in the regulation of p300 expression by cortisol in human amnion fibroblasts. (**A**) Blockade of cortisol (1 µM; 24 h)-induced *EP300* (encoding p300) mRNA expression by CREB inhibitor 666-15 (10 μM). n = 4. (**B**) Ineffectiveness of STAT3 inhibitor S3I-201 (10 μM) (B, n = 4) on the induction of *EP300* mRNA expression by cortisol (1 µM; 24 h). (**C**) Diagrams illustrating the putative binding sites of CREB in *EP300* gene promoter. Brown box indicates CREB binding sites. Arrows indicate primer aligning positions in ChIP assay. P1, forward primer; P2, reverse primer; TSS, transcription start site. (**D**) ChIP assay showing increased CREB enrichment at *EP300* gene promoter in human amnion fibroblasts with cortisol treatment (1 μM; 12 h). n = 4. IgG served as negative control. (**E**,**F**) Blockade of cortisol (1 µM; 24 h)-induced *EP300* mRNA expression by EP2 receptor antagonist PF-04418948 (PF; 10 μM; **E**) and PKA inhibitor PKI 14-22 amide (PKI; 5 μM; F). n = 4. (**G**) Induction of *EP300* mRNA expression by PGE2 (1 µM; 6 h). n = 3. (**H**,**I**) Blockade of PGE2 (1 µM; 6 h)-induced *EP300* mRNA expression by EP2 receptor antagonist PF-04418948 (PF; 10 μM; **H**) and PKA inhibitor PKI 14-22 amide (PKI; 5 μM; **I**). n = 4. Data are represented as means + SEM. Statistical analysis was performed with paired Student’s *t*-test (**D**,**G**) or one-way ANOVA test (A, B, E, F, H, and I) followed by Newman–Keuls multiple comparisons test. * *p* < 0.05, ** *p* < 0.01, *** *p* < 0.001 vs. control group. # *p* < 0.05, ## *p* < 0.01, ### *p* < 0.001 vs. cortisol- or PGE2-treated group.

**Figure 7 ijms-24-10881-f007:**
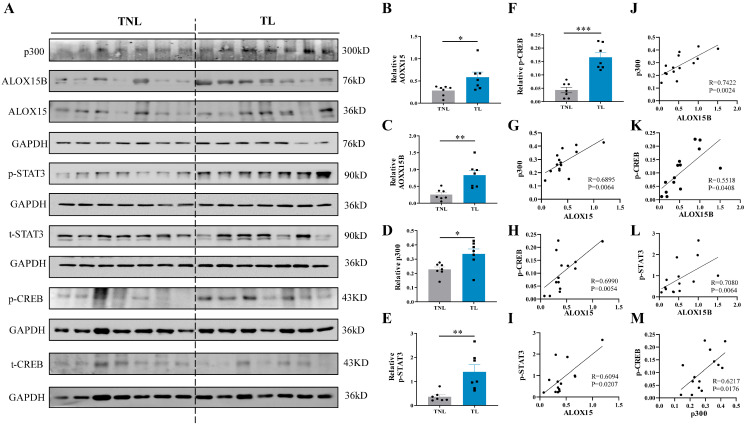
Increased abundance of ALOX15, ALOX15B, p300, p-CREB, and p-STAT3 in the human amnion at parturition. (**A**–**F**) Abundance of ALOX15, ALOX15B, p300, p-CREB, and p-STAT3 in the human amnion collected from term delivery with labor (TL; n = 7) and without labor (TNL; n = 7) as measured with Western blotting. (**G**–**I**) Pearson analysis showing positive correlation between ALOX15 and p300 (**G**), p-CREB (H), or p-STAT3 (**I**) abundance in the human amnion (n = 14). (**J**–**L**) Pearson analysis showing positive correlation between ALOX15B and p300 (J), p-CREB (K), or p-STAT3 (**L**) abundance in the human amnion (n = 14). (**M**) Pearson analysis showing positive correlation between p300 and p-CREB in the human amnion (n = 14). Statistical analysis was performed with unpaired Student’s *t*-test (**C**–**F**) or Mann–Whitney U test (B). * *p* < 0.05, ** *p* < 0.01, *** *p* < 0.001 vs. TNL.

**Figure 8 ijms-24-10881-f008:**
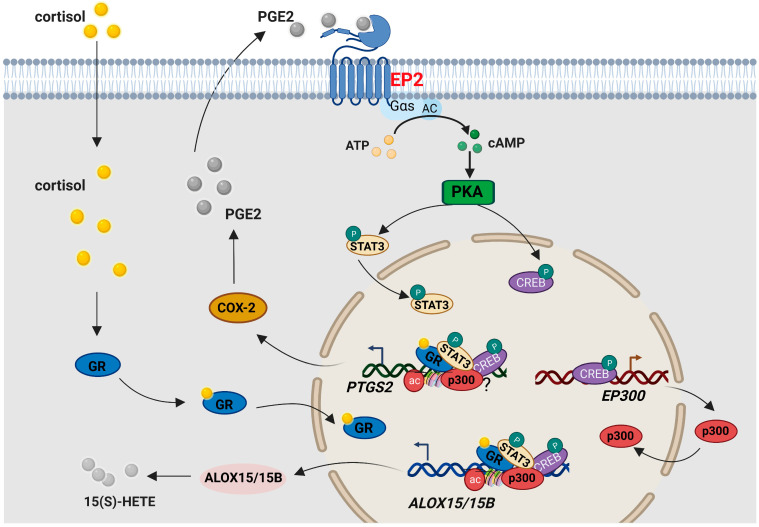
Diagram illustrating the pathway underlying the induction of ALOX15/15B expression by cortisol in human amnion fibroblasts. By binding to EP2 receptors, PGE2 activates the cAMP/PKA pathway, followed by CREB and STAT3 phosphorylation. Phosphorylated CREB and STAT3 not only induce *PTGS2* (encoding COX-2) but also *ALOX15/15B* expression. PGE2 produced by COX-2 further reinforced this feed-forward loop. In addition, phosphorylated CREB also induces p300 expression. Increased p300 not only leads to H3K27 acetylation at *ALOX15/15B* gene promoters but also interacts with phosphorylated CREB and STAT3 at *ALOX15/15B* gene promoters to induce ALOX15/15B expression and 15(S)-HETE production. When cortisol is present, GR activated by cortisol interacts with phosphorylated CREB and STAT3 to further stimulate *PTGS2* and *ALOX15/15B* expression. Gαs, stimulatory G protein; AC, adenylate cyclase; ATP, adenosine 5′-triphosphate; cAMP, cyclic adenosine monophosphate.

## Data Availability

The transcriptomic sequencing data of the amnion fibroblasts with or without cortisol treatment have been submitted to the GEO data repository (GSE166320). The original data and materials presented in the study are available from the corresponding authors upon reasonable request.

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
