# Peer review of "Paradoxical Induction of ALOX15/15B by Cortisol in Human Amnion Fibroblasts: Implications for Inflammatory Responses of the Fetal Membranes at Parturition"

_ijms, 2023, doi:10.3390/ijms241310881_

Round 1
Reviewer 1 Report
Summary: This well-written paper is one in a series of papers from this group on the role of cortisol regeneration by amnion fibroblasts of the fetal membranes in the process of parturition. Contrary to conventional assumptions, the authors suggest cortisol, rather than acting as a prototypical anti-inflammatory steriod, actually is proinflammatory in amnion fibroblasts. This novel idea, though controversial, has been supported by the present work and by previous papers from their lab.
This paper, in particular, shows that cortisol induces the expression of 15-lipoxygenase-1, one of two separate lipoxygenase enzymes involved in the synthesis of hydroxylated lipid derivatives from arachidonic acid. In the case of amnion fibroblasts and most other cells, 15-lipoxygenase-1 produces 15-hydroxyeicosatetraenoic acid (15-HETE), that has many biological functions. The authors suggest that, in the case of amnion fibroblasts, 15-HETE plays a pro-inflammatory role. There is also evidence in the literature that 15-HETE can serve as a substrate for the synthesis of anti-inflammatory/pro-resolving lipid mediators. It would be of interest to know the biological effects of 15-HETE on amnion cells. For example, the 15-HETE regulate the synthesis of proinflammatory cytokines, matrix metalloproteinases or induce COX-2? That would provide a more persuasive link to the onset of parturition.
The paper is worthy of publication; however, the ultimate interpretation of 15-LOX-1-derived 15S-HETE will await further functional studies.
Author Response
This well-written paper is one in a series of papers from this group on the role of cortisol regeneration by amnion fibroblasts of the fetal membranes in the process of parturition. Contrary to conventional assumptions, the authors suggest cortisol, rather than acting as a prototypical anti-inflammatory steriod, actually is proinflammatory in amnion fibroblasts. This novel idea, though controversial, has been supported by the present work and by previous papers from their lab.
This paper, in particular, shows that cortisol induces the expression of 15-lipoxygenase-1, one of two separate lipoxygenase enzymes involved in the synthesis of hydroxylated lipid derivatives from arachidonic acid. In the case of amnion fibroblasts and most other cells, 15-lipoxygenase-1 produces 15-hydroxyeicosatetraenoic acid (15-HETE), that has many biological functions. The authors suggest that, in the case of amnion fibroblasts, 15-HETE plays a pro-inflammatory role. There is also evidence in the literature that 15-HETE can serve as a substrate for the synthesis of anti-inflammatory/pro-resolving lipid mediators. It would be of interest to know the biological effects of 15-HETE on amnion cells. For example, the 15-HETE regulate the synthesis of proinflammatory cytokines, matrix metalloproteinases or induce COX-2? That would provide a more persuasive link to the onset of parturition.
The paper is worthy of publication; however, the ultimate interpretation of 15-LOX-1-derived 15S-HETE will await further functional studies.
Reply: We sincerely appreciate your positive and constructive comments on our manuscript. As a matter of fact, we have demonstrated that 15(S)-HETE can potentiate pro-inflammatory mediators-induced COX-2 expression and PGE2 production in human amnion fibroblasts in our previous study (J. Lipid Res. 2022. 63(11): 100294), indicating 15(S)-HETE plays a crucial role in the inflammation reaction of the human amnion. However, we are not clear whether 15(S)-HETE regulates the synthesis of pro-inflammatory cytokines and matrix metalloproteinases in the amnion, which need further investigation in the future.
Reviewer 2 Report
In this manuscript, Zhang et al., address a very interesting subject and explain very clearly the findings obtained regarding the paradoxical induction of ALOX15/15B when fibroblasts are treated with cortisol. The methods are very well described and the results are presented very clearly.
I would only have some questions and minor comments that would be worth clarifying in order to strengthen the manuscript:
What is the cortisol concentration that can be reached in the amnion layer of human fetal membranes? Or what was the argument for selecting the 0.01-1 μM concentration of cortisol to assess the expression of ALOX15/ALOX15B?
In the discussions, it would be useful to mention the limitations of the study or future approaches, such as the silencing of receptors and/or proteins involved in the proposed signaling pathway, since only pharmacological blockades with antagonists were used to elucidate the pathway.
Line 483: correct to μg
Author Response
In this manuscript, Zhang et al., address a very interesting subject and explain very clearly the findings obtained regarding the paradoxical induction of ALOX15/15B when fibroblasts are treated with cortisol. The methods are very well described and the results are presented very clearly.
Reply: Thank you very much for your support and constructive comments on our manuscript.
I would only have some questions and minor comments that would be worth clarifying in order to strengthen the manuscript:
What is the cortisol concentration that can be reached in the amnion layer of human fetal membranes? Or what was the argument for selecting the 0.01-1 μM concentration of cortisol to assess the expression of ALOX15/ALOX15B?
Reply: Thank you for raising this important issue. Our previous and present studies have revealed that the cortisol concentration in the amnion can reach up to around 5 μM at parturition. That is why we used 0.01-1 μM in this study, which is physiologically relevant.
In the discussions, it would be useful to mention the limitations of the study or future approaches, such as the silencing of receptors and/or proteins involved in the proposed signaling pathway, since only pharmacological blockades with antagonists were used to elucidate the pathway.
Reply: Thank you for your suggestions. We have included this limitation in the discussion section.
Line 483: correct to μg
Reply: Corrected.